# Use of Thermochromic Properties of VO$_2$ for Reconfigurable Frequency Selection

Qassim Abdullahi [1,*], Adrian Dzipalski [2], Clement Raguenes [1], Nelson Sepulveda [3], Gin Jose [4], Atif Shanim [5], George Goussetis [1], Duncan Hand [2] and Dimitris E. Anagnostou [1]

1 Institute of Signals, Sensors and Systems, Heriot-Watt University, Edinburgh EH14 4AS, UK
2 Institute of Photonics and Quantum Sciences, Heriot-Watt University, Edinburgh EH14 4AS, UK
3 Department of Electrical and Computer Engineering, Michigan State University, East Lansing, MI 48824, USA
4 School of Chemical and Process Engineering, University of Leeds, Leeds LS2 9JT, UK
5 Electrical and Computer Engineering, King Abdullah University of Science and Technology, Thuwal 23955, Saudi Arabia
* Correspondence: qsa1@hw.ac.uk

**Abstract:** The thermochromic nature of vanadium dioxide (VO$_2$) has facilitated many promising applications for reconfigurable frequency selectivity. The phase-changing property of VO$_2$ was used to realise a reconfigurable frequency-selective surface (FSS) capable of manipulating electromagnetic waves for different functionalities. Diffractive optical elements (DOE) are used for diffracting laser beams to form conductive FSS images on the VO$_2$ wafer for frequency selectivity. The dipoles on the VO$_2$ wafer generate a stop band response of 12 dB and 10 dB for unit cells of the single dipole and double dipole at 3.5 GHz, respectively. A 10 GHz FSS array is projected by DOE on the 2-inch VO$_2$ wafer with a filtering effect of 13 dB at 9.5–10.5 GHz. This solution is used to design a radar cross-section (RCS) modification FSS with reflected waves of about 20 dB higher reflectivity in the backscattering direction than in the specular direction.

**Keywords:** frequency selective surfaces; reconfigurable components; vanadium dioxide

## 1. Introduction

Vanadium dioxide (VO$_2$) exhibits thermochromic properties during the transition as its optical, electrical, and thermal conductivity properties change. This colour change is due to a change in its crystalline phase and structure from a monoclinic to rutile crystal, along with the change in electrical properties such as resistivity during the insulator-to-metal transition (IMT) [1,2]. Thermochromic materials such as oxides of transition metal elements used in buildings are known for being good solar radiation reflectors [3]. Thermochromic materials are also suitable retroreflective materials as they can reflect incoming radiation waves toward the initial incident direction [3]. The need for manipulation of electromagnetic (EM) waves has paved the way for the development and use of metasurfaces and frequency selective surfaces (FSS) to fulfil ground-breaking leads in applications such as RCS reduction [4], cloaking [5], optical illusion [6], and negative refraction [7,8]. Work on VO$_2$-based reconfigurable antenna and microwave devices has also been interesting [9–13]. Using IMT, the switching mechanism can tune the operating frequency, configure the structural patterns, and manipulate the electric size of the devices by changing the phase [14–16].

A new area of interest has recently arisen that aims to digitise metasurfaces and FSS applications. These new concepts propose systems that combine the physical metamaterial structures to a digital world of Information Technology by using control sequences to manipulate EM waves through metasurfaces and FSS of different functionalities. Pursuing such systems has led to the development of low-weight metasurfaces and FSS, which have become easier to design and fabricate than their traditional counterparts [17]. These planar structures, accompanied by their information-feeding network, have led to innovative

devices that expertly manipulate EM waves [18–20]; however, these designs, limited by their fabrication, operate within specific tuneable functionalities, which, according to [21], hinders practical applications.

Recently, developments in tuneable metasurfaces and FSS based on phase change materials such as chalcogenide glasses (GST) [22], liquid crystals [23], and graphene [23–26] have been researched; however, some of the limitations to these phase change materials (PCM) include volume expansion, surface oxidation, slow actuation, and high laser pulse that exceed the laser damage threshold of the surrounding materials [27].

The use of $VO_2$ mitigates such limitations, making it an attractive phase change material. Its transition can be triggered thermally at about 68 °C, which makes it versatile and easy to control [28]. Additionally, the transition temperature in $VO_2$ is lower than that of other PCMs and has more switch cycle stability [29–31]. This transition temperature depends on the dopants present during a deposition and the type of deposition technique used [32]. $VO_2$ also undergoes a natural reverse transition to its initial state without excitation [9]; therefore, it can be easily integrated with materials of low-temperature thresholds, such as capacitors and sensing resistors in $VO_2$ switching application circuitry. The low transition temperature makes it very attractive for FSS and reconfigurable microwave devices, which require adaptable frequency bands, and variable radiation polarisation and patterns [33].

Metasurfaces and FSS that use $VO_2$ as a phase-changing material have been reported in several works [33–45]. Among them are thin film metasurfaces that utilise $VO_2$ phase transition through optical switching at nanoscales to produce adjustable permittivity to achieve functionalities such as perfect absorption [38], minimising reflection [39], and optical diodes [17]. In these designs, metallic-based metasurfaces suffer lower efficiencies because of absorption losses and weaker magnetic resonances than $VO_2$-dielectric metasurfaces [27].

Electrically triggered $VO_2$-based metasurfaces have been explored for their useful concept in real-world applications. One relevant work is [41], which first reported using a planar electrode array on split-ring resonators to tune resonance. In [42], a $VO_2$ thin film was sandwiched between metal antennas and a dielectric substrate. The top layer consisted of an antenna array with unit cells connected, enabling electrical current to flow through the structure to trigger the IMT of $VO_2$ for optical reflectance modulation electrically. In [43], nano-sized $VO_2$ elements were introduced at the feed gap of a bowtie antenna, while [44] used joule heating on a reflect array architecture with a $VO_2$ layer for spectral tuning and intensity modulation; however, electric control of the IMT in $VO_2$ is a complex and expensive process due to the integration of electrodes with $VO_2$ during the etching process that uses highly reactive wet-etchant chemicals such as $HNO_3$ [37,42,43,45]. Moreover, in optical switching applications, the presence of such electrodes will introduce distortions in the optical properties [37]. Using external means of switching, such as holographic storage or lasers for optically imprinting reconfigurable patterns, takes advantage of the reversible transition of $VO_2$ thin films, which do not have the same limitations and constraints as previously presented [45,46].

Several studies have employed $VO_2$ thin film for FSS and spatial filter applications in the THz region [47–51] and RF region [52–54]; however, these designs consist of metallic resonators and slots with switchable $VO_2$ layers limited to dual states. In this work, we present a dynamically reconfigurable FSS and spatial filter capable of operating at a wide range of frequencies and states, limited only to the number of unit cells that can fit onto the film surface. We use DOE to form conductive patterns on a purely formed $VO_2$ thin film and measure the transmission characteristics. This concept does not require complex etching techniques to amalgamate metallic layers with the $VO_2$. The work investigates the transition behaviour of $VO_2$ to demonstrate optically actuated unit cells in the 2.5–5 GHz bands and FSS array at 10 GHz.

## 2. Materials and Methods

### 2.1. Measurement Setup

The measurement setup to form conductive patterns on the VO$_2$ wafer and measure its transmission characteristics consisted of a free space arrangement with two broadband horn antennas over the frequency ranges of 2.5–5 GHz and 8–10 GHz and a metallic wall with a circular aperture of 5.08 cm diameter that acted as the sample holder. The setup dimensions are defined in [55] based on a half-power beamwidth to ensure the EM waves pass through the aperture. The distance between the antenna and sample, r, was more significant than the wavelength, λ = 0.14 m at 2 GHz, and the aperture.

A gated reflection load (G.R.L.) calibration was used with a metal plate containing an aperture to confine the resolution to the sample; two primary calibration standards are considered here. They include VNA calibration and free space calibration [56].

The network analyser calibration involved using short, open, load, and through (SOLT) standards that removed unwanted effects of cables and adaptors by shifting the plane of reference to the antennas. The calibration was performed at room temperature by using the Keysight 85052D calibration kit due to its broadband nature. Initially, this calibration kit works with coaxial setups; however, short, open, and load standards were inserted into the antenna connectors, which were connected for the through standard. The path losses were thus considered in the calibration. The remaining reflections, in the antennas and on the metal plate globally, were handled through post-process time-gating. The gate was defined by comparing the time domain responses of the empty fixture with the metal in the fixture: the only difference between them was due to the sample difference (Cu vs. air). All the other areas were filled by unwanted reflections that were thus suppressed. N5225A PNA from Agilent was used for all measurements, connected to two horn antennas through 3.5 mm connectors, as shown in Figure 1a,b.

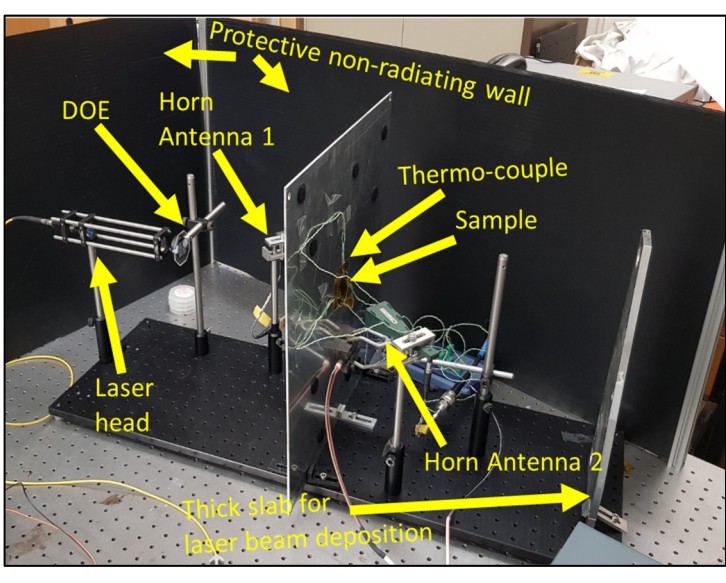

(**a**)

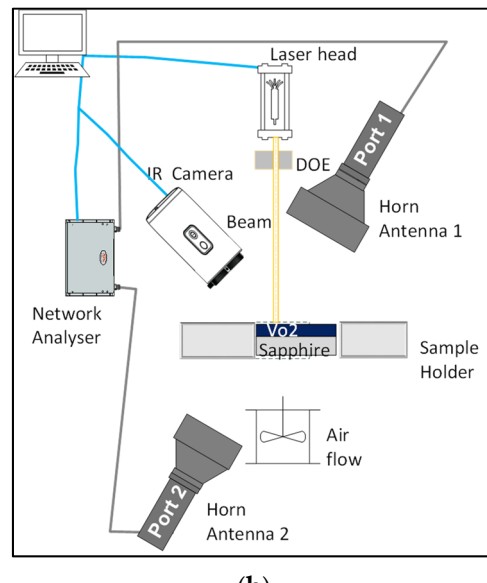

(**b**)

**Figure 1.** *Cont.*

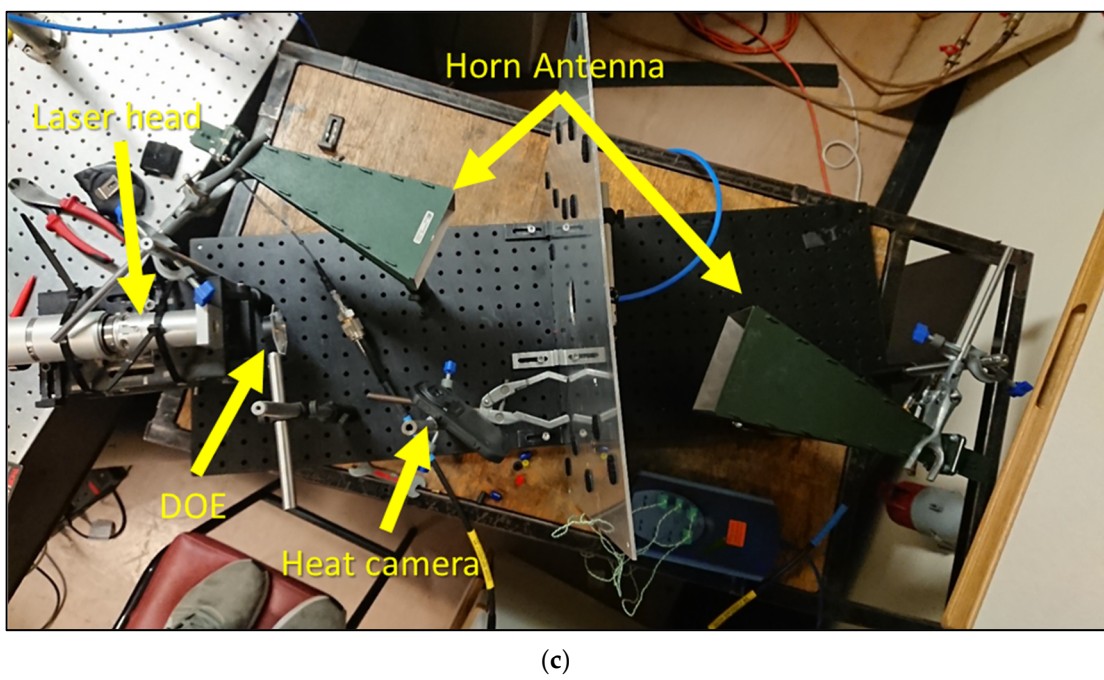

(**c**)

**Figure 1.** (**a**) Experimental setup in the lab. (**b**) Schematic of the measurement setup. (**c**) Elevated view of experimental setup in the lab.

### 2.2. Phase Pattern Formation

The DOE is designed to operate in the Fresnel regime [57] using an iterative Fourier transform algorithm (IFTA) to generate the phase profiles. The algorithm shown in Figure 2a finds the phase of a propagating optical field between two parallel planes with known amplitudes of the fields [58]. The derived phase is then used to calculate the modulation function of the phase to produce a known far-field distribution from the illuminating beam. IFTA takes the amplitude of the image and the Fourier transform of the image (amplitude of the diffraction plane) as its inputs. The process recovers phase information of the Fourier transform and reconstructs the signal. At each iteration, the process alternates between the frequency and time domains using the input amplitudes to improve phase estimation [59,60].

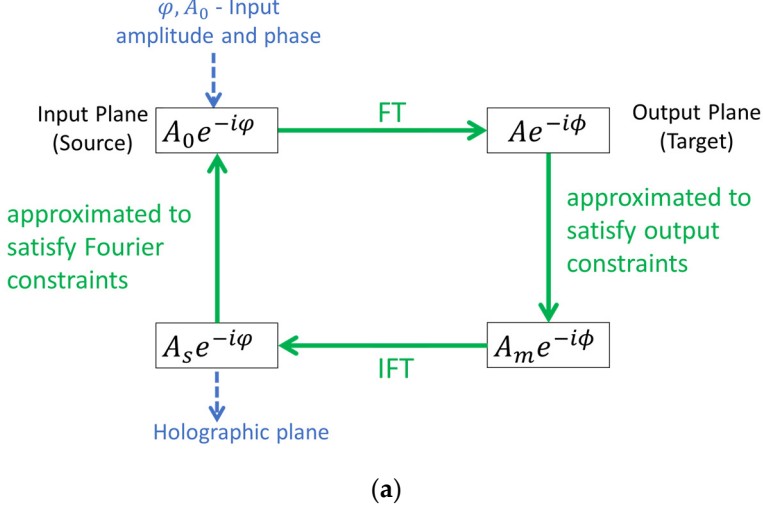

(**a**)

**Figure 2.** *Cont.*

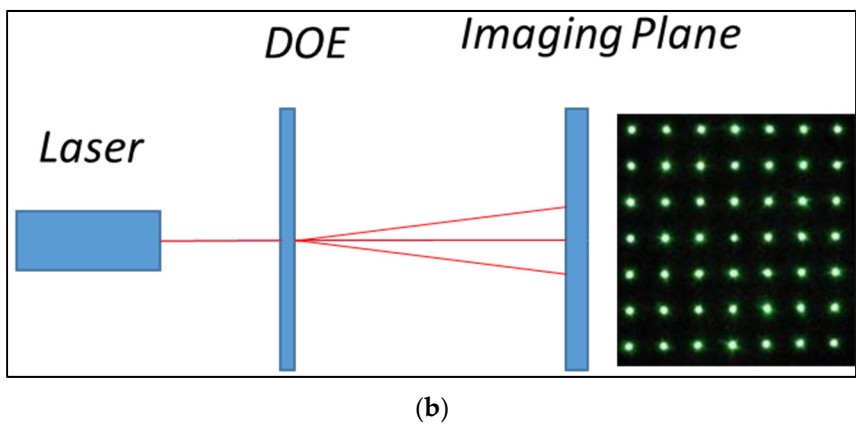

**(b)**

**Figure 2.** (**a**) Flowchart of DOE formation through phase retrieval. $\varphi$, $A_0$ are estimates of the image phase and amplitude respectively. The Fourier transform of image plane, $A_0e^{-i\varphi}$ is used to recover the output plane, $Ae^{-i\phi}$. The output plane calculated from the diffraction pattern is estimated to the desired Fourier transform output intensity, $A_me^{-i\phi}$. The estimate of the Fourier transform is then inverse Fourier transformed to $A_se^{-i\phi}$. The IFT signal must is changed if the new estimate of the transform does not satisfies the desired output constraints then the whole process is repeated until both the Fourier constraint and output constraint are satisfied. (**b**) Schematic of phase pattern formation with DOE.

The DOE was fabricated at the class-1000 cleanroom nanofabrication centre of the Heriot-Watt University through photolithography on a substrate of 1 mm thickness through reactive ion etching (RIE).

The 100 W EPS-Z SPI laser system used a maximum pulse energy of 0.7 mJ, a duration of 490 ns, and a repletion rate of 28 kHz for etching. The etch depth was optimised for 1064 nm with a minimum feature size of 5 $\mu$m and aperture of 1 mm for an optical output of a $1 \times 32$ array. The projected dipole size was augmented by changing the propagation distance between the DOE and imaging plane, as described in Figure 2b.

Figure 3a–c show the optical diffraction pattern of dipoles on the VO$_2$ film at specified propagation distances, while Figure 3d–f show the thermal actuating laser beam diffraction pattern of the dipoles. The dipole designs consisted of unit cells at 3, 3.3, and 4 GHz projected on the VO$_2$ film circular surface of 50.8 mm diameter. The unit cell designs were selected at lower frequencies to obtain larger dipoles due to undesirable signal rejections caused by the heat spread on the VO$_2$ wafer because of the changing thermal conductivity. The propagation distances used were 57.5 cm for 39 mm by 2.9 mm at 3 GHz, 42.5 cm for 30 mm by 2.9 mm at 3.3 GHz, and 32.5 cm for 23 mm by 2.9 mm at 4 GHz.

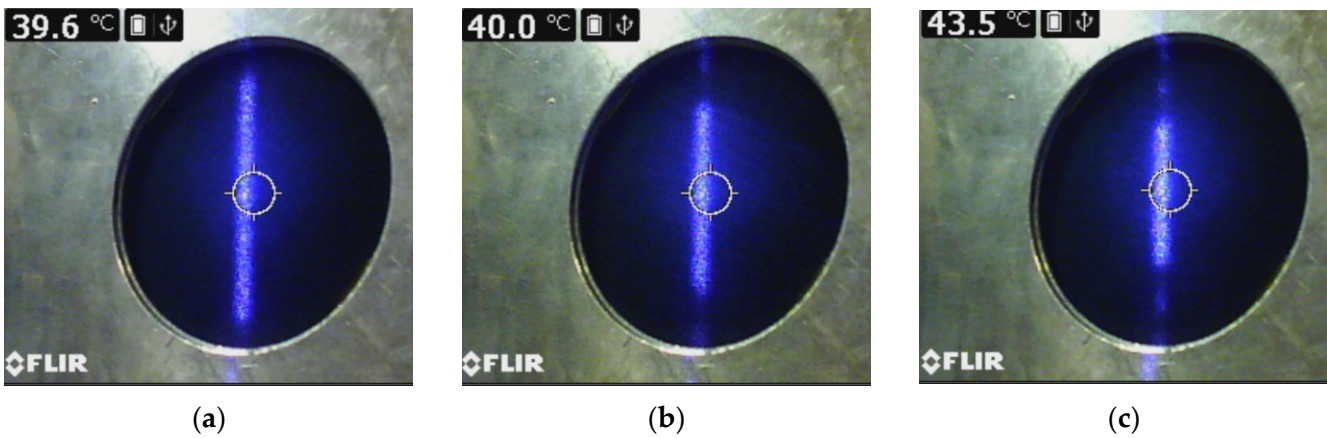

**(a)**          **(b)**          **(c)**

**Figure 3.** *Cont.*

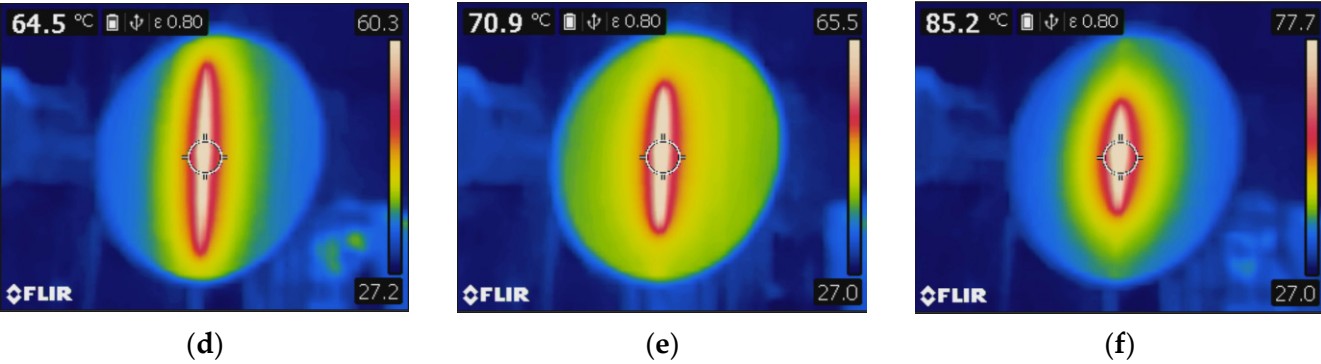

**Figure 3.** The dipole dimensions at varying propagation distances: (**a**) Infrared image of the diffraction pattern for 57.5 cm for a 39 × 2.9 mm dipole; (**b**) Infrared image of the diffraction pattern for 42.5 cm for a 30 × 2.9 mm dipole; (**c**) Infrared image of the diffraction pattern for 32.5 cm for a 23 × 2.9 mm dipole; (**d**) thermal image with laser for (**a**); (**e**) thermal image with laser for (**b**); (**f**) thermal image with laser for diffraction pattern for 32.5 cm for a 23 × 2.9 mm dipole.

## 3. Results

### 3.1. Material Thermal Property

In FSS applications, the individual elements are of the order of the operating wavelength $\lambda/2$, which resonates at the design frequency [61]. FSSs have periodicity equal to half the resonant frequency wavelength [62]. Any change in element dimensions will significantly affect the desired frequency response. The thermal conductivity of the $VO_2$ wafer changes alongside the electrical conductivity during the transition, becoming more conductive, resulting in accumulated thermal profile changes [63–65]; therefore, it is essential to achieve good heat confinement to maintain the pattern of the projected DOE image on the wafer.

The film in Figure 4a–c consists of a 400 nm $VO_2$ layer deposited on a 2-inch circular substrate of sapphire 275 μm thick. The wafer contains $VO_2$ thin layer on a sapphire substrate. The $VO_2$ layer has a thermal conductivity (K) of 0.9 W/mK at room temperature that increased to 2.5 W/mK after transition, specific heat capacity (c) of 656.31 J/KgK, and a density (ρ) of 4.58 g/cm$^3$. The sapphire layer has a thermal conductivity of 23.1 W/mK, a specific heat capacity (c) of 761 J/KgK, and a density (ρ) of 3.98 g/cm$^3$ as seen in Figure 4a.

The heat confinement of the dipole patterns is difficult to maintain at higher frequencies with smaller interelement spacing between unit cells due to the thermal conductivity associated with the phase transition. The confinement should have enough temperature difference between the dipoles of the FSS pattern and the interelement space between them such that only the dipole pattern attains the transition temperature. The heat confinement is improved by designing the unit cell with larger interelement spacing between the dipole elements and fewer elements.

The wafer is cooled from the back to improve the thermal confinement of the diffracted image, as shown in Figure 5. The cooling process transfers the heat transfer coefficient of the sample to the air surrounding it, thereby reducing the temperature in the sample. The airflow required to dissipate the heat is approximated when the density of the air is known [66]. The wafer temperature is monitored from the front with the FLIR E5-XT thermal imaging camera alongside K-type thermocouples at the back. The temperature distribution on the wafer profile is obtained from the thermal mapping of the camera, as shown in Figures 3 and 6.

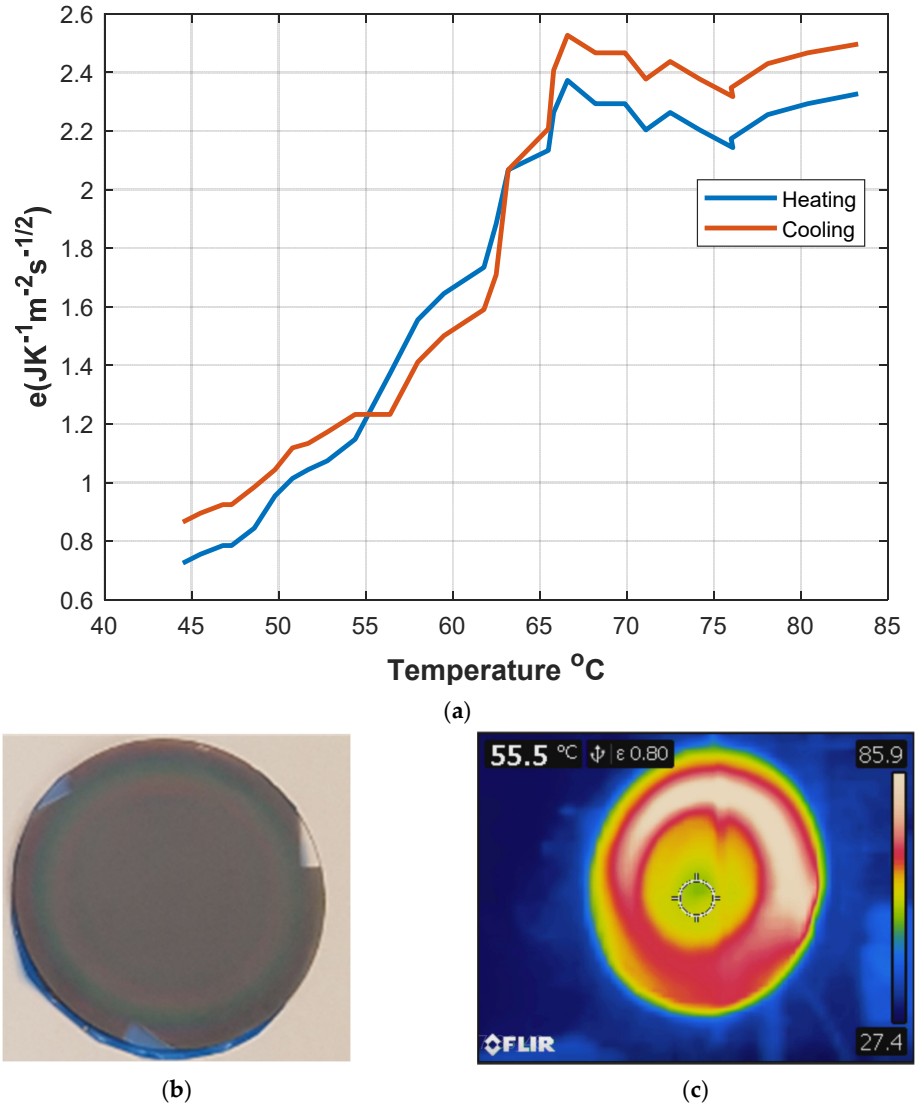

(a)

(b)

(c)

**Figure 4.** (**a**) Estimated thermal effusivity of the wafer containing a 2-inch circular $VO_2$ thin layer with 400 nm thickness on 275 μm sapphire substrate. (**b**) $VO_2$ sample. (**c**) The thermal infrared image is taken from the FLIR camera of the $VO_2$ sample.

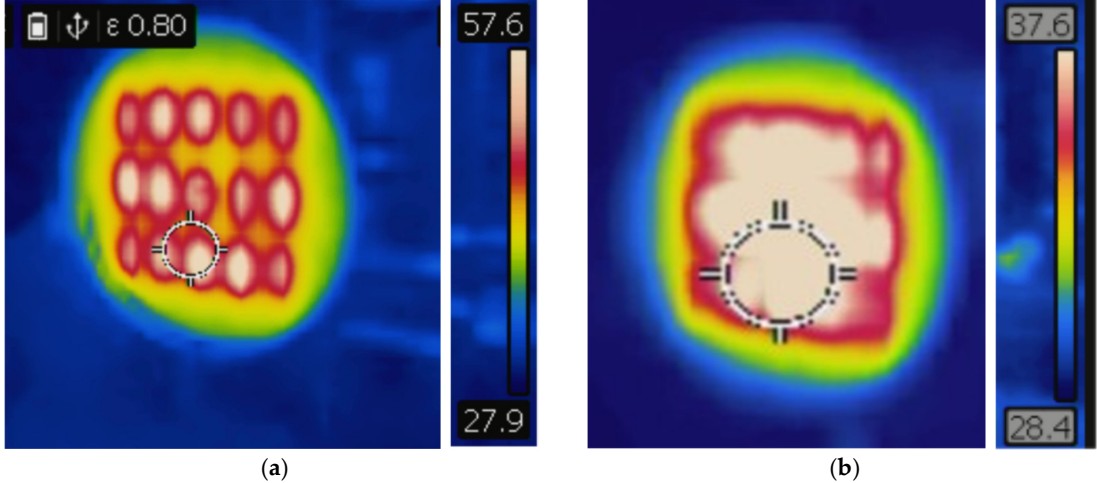

(a)

(b)

**Figure 5.** Thermal image of dipole array through Pulse Wave: (**a**) with cooling; (**b**) no cooling.

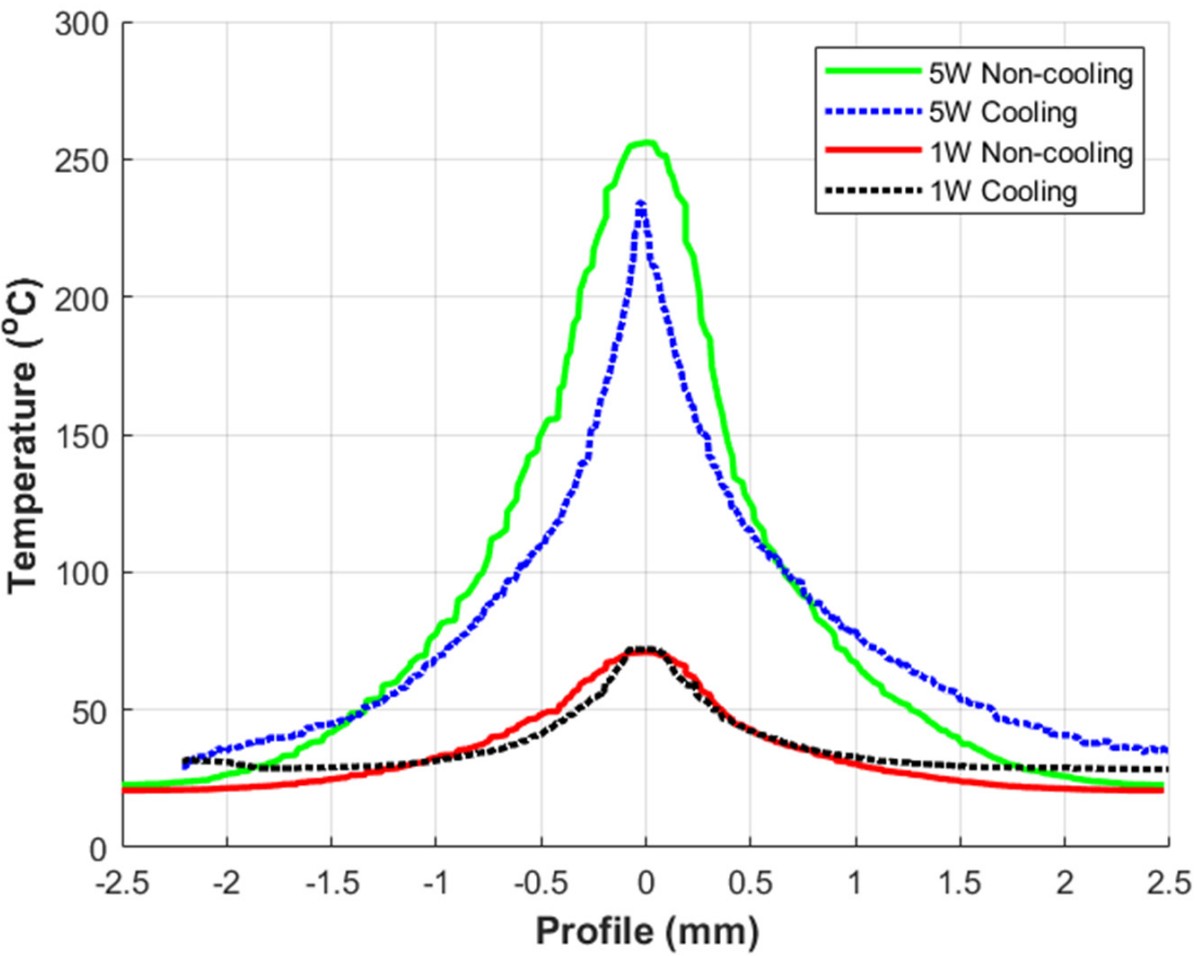

**Figure 6.** The temperature distribution of unit cells on the sapphire wafer with and without cooling.

A pulsed wave (P.W.) laser scheme is used to heat the sample to the transition temperature. Before transition, the heat profile associated with the laser heating at transition temperature is less than 8 mm. On approaching the transition temperature, the heating profile expands to 8 mm. These changes are due to the increase in thermal conductivity during the IMT transition. The DOE image pattern in Figure 5 is unit cells modelled after FSS dipole surface patterns at 10 GHz. The proposed design is expected to give a passband response at the specific design frequencies.

Without cooling, undesirable signal rejections are obtained due to the heat spread on the VO$_2$ wafer from the changing thermal conductivity, as shown in Figure 6. Figure 6 shows that heat expansion will not significantly alter the shape of the dipole at 110 °C. The heat expansion is attributed to the thermal conductivity of the base wafer on which the thin film is deposited. A base wafer with a lower thermal conductivity will be better suited for this application.

### 3.2. Microwave Properties

Dipole FSS arrays consist of unit cells periodically distributed on a substrate or a metallic sheet. These unit cells function as a radiation filter to manipulate waves [62].

VO$_2$ thin film has demonstrated thermochromic properties as it undergoes IMT during the temperature rise. As the temperature increases, the transmission coefficient decreases while its reflective properties rise, as shown in Figure 7. This decrease is due to the rise in electrical conductivity. The results in Figure 7 show the entire film being heated from 20 °C to 110 °C, and a 16 dB change in transmission coefficient is observed. This property

makes for an attractive reconfigurable FSS that is achieved by defocusing the beam to obtain various dipole patterns and using a diffuser to ensure uniform illumination.

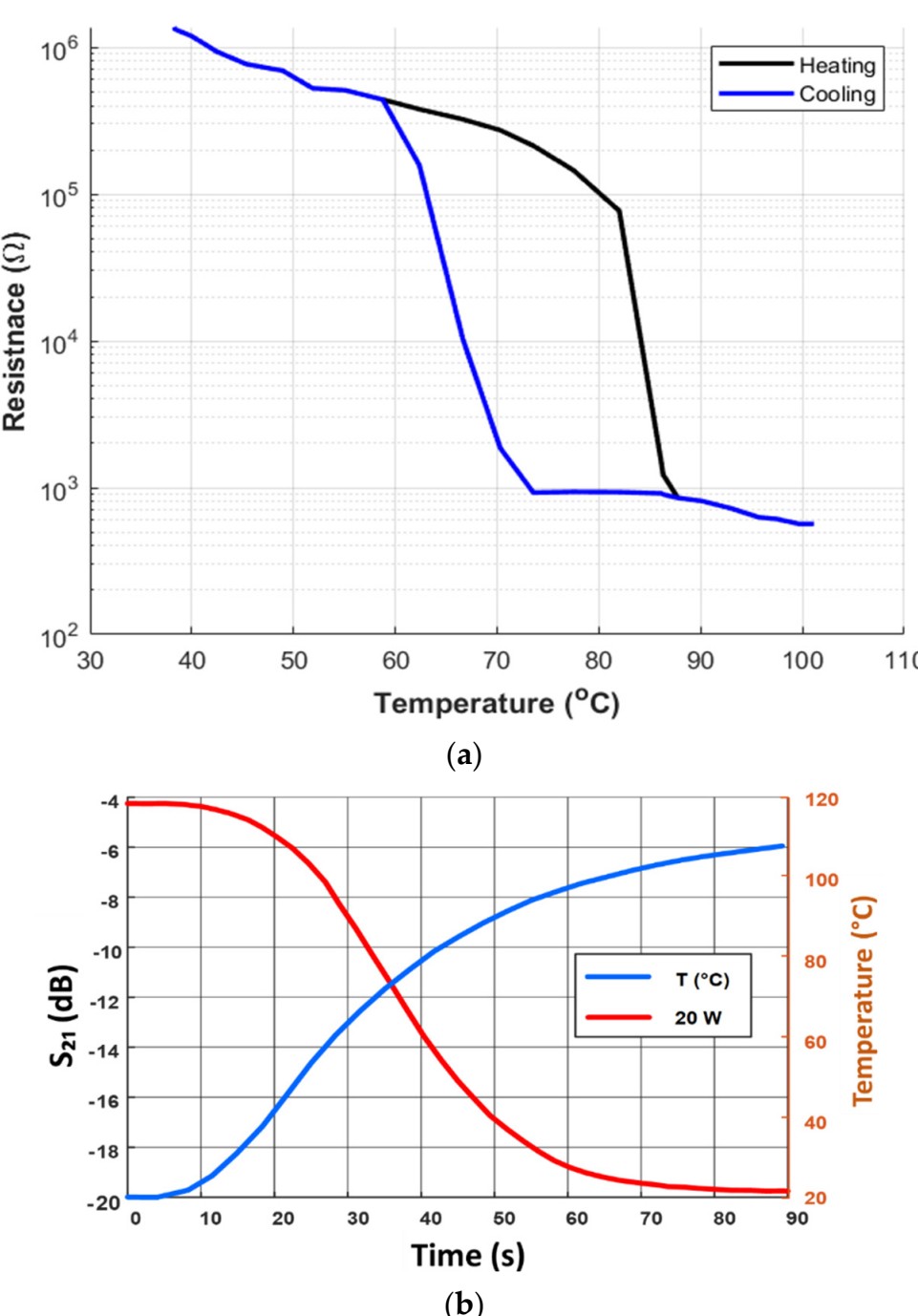

**Figure 7.** (**a**) Measured electrical resistance of VO$_2$ thin film during the heating and cooling phase. (**b**) Measured transmission characteristics of full wafer heating attenuated by 16 dB at 10 GHz.

### 3.2.1. Single Dipole Unit Cell

A single unit cell dipole is illuminated on the VO$_2$ wafer matching a Cu unit cell of similar dimension. The length of the dipoles varies, as shown in Figure 8a,b. In Figure 9, the transmission frequency response of the VO$_2$ wafer for dipoles of dimensions 30 mm × 2.9 mm at 3.3 GHz and 39 mm × 2.9 mm at 3 GHz are compared with Cu material of the exact dimensions on the same substrate.

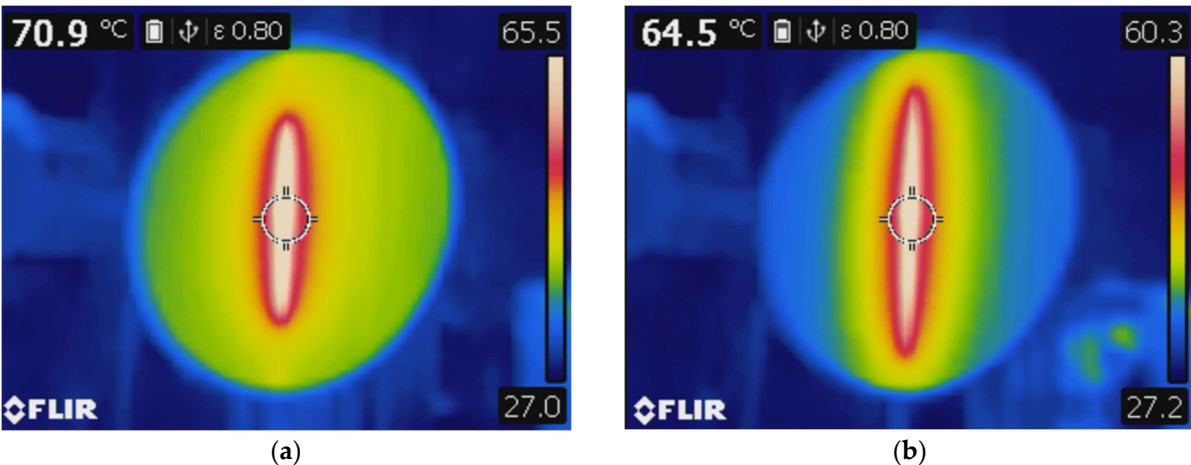

**Figure 8.** Captured images of laser projected single dipole pattern with dimensions: (**a**) 30 mm × 2.9 mm (~3.3 GHz); (**b**) 39 mm × 2.9 mm (~3 GHz).

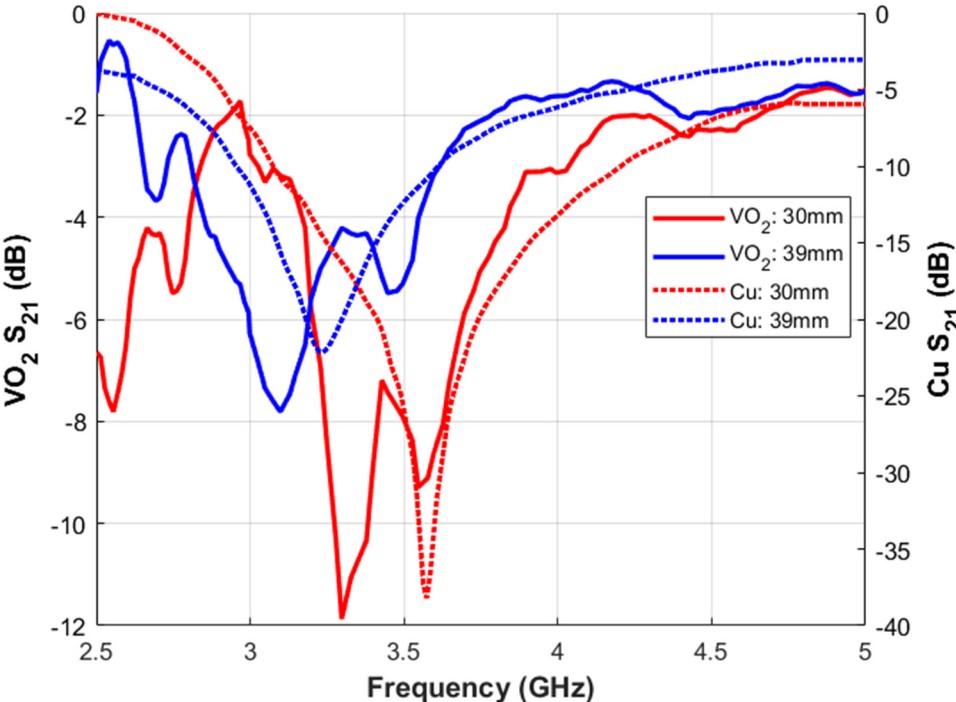

**Figure 9.** The graph of the measured transmission coefficient of single dipole element of VO$_2$ and Cu for lengths 30 and 39 mm.

In Figure 9, the 30 mm by 3 mm pattern heated to 110 °C generates a filtering frequency response with resonance slightly shifted to 3.4 GHz at 12 dB compared to the Cu dipole with a resonance at 3.6 GHz at 37 dB. The 39 mm by 3 mm pattern generates a filtering response of 7 dB at 3.1 GHz, while its Cu counterpart has a 21 dB response at 3.3 GHz. In both comparisons, the magnitude of the VO$_2$ dipoles is three times less than the Cu dipoles, primarily due to the electrical conductivity of Cu. It washes away the diffracted pattern, which affects the transmission frequency response obtained due to the rise in thermal conductivity of the VO$_2$ layer on going through IMT, although, by removing cooling, higher temperatures can be achieved.

### 3.2.2. Two (Double) Dipole Unit Cell

Double dipole patterns were imprinted on the film, as shown in Figure 10. The dipoles are equal in length, L, and width, W, with a spacing, D, as shown in Figure 10 (inset). D varies from 5 mm to 30 mm compared to its Cu counterpart.

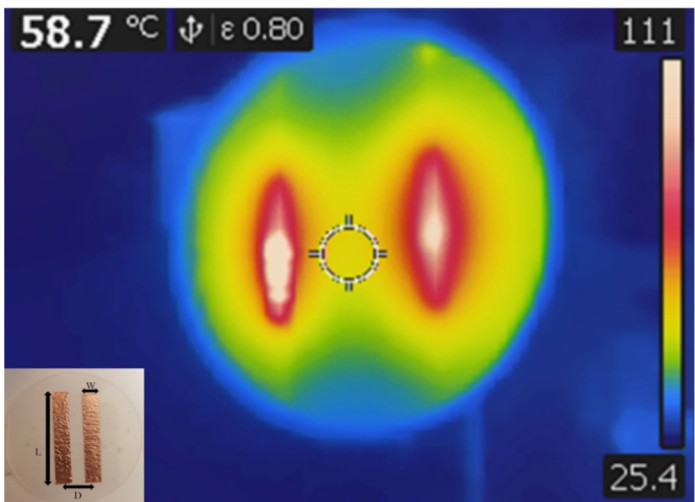

**Figure 10.** Captured laser image of thermal profile of double dipole pattern on VO$_2$ over sapphire wafer during heating. Inset: geometry of double dipoles showing the length, L, width, W, and spacing, D.

The magnitude of the transmission coefficient (S$_{21}$) for VO$_2$ double dipoles at different spacing, D, is shown in Figure 11. The frequency response of the dipoles stabilises by varying D with a fixed L and W of 39 mm and 3 mm, respectively. The resonance varied slightly from 3.2 to 3.4 GHz and became more defined with an increasing magnitude as the separation distance increased. This variance is due to the diffracted pattern moving closer to the designed dimensions and heat expansion caused by the changing thermal conductivity.

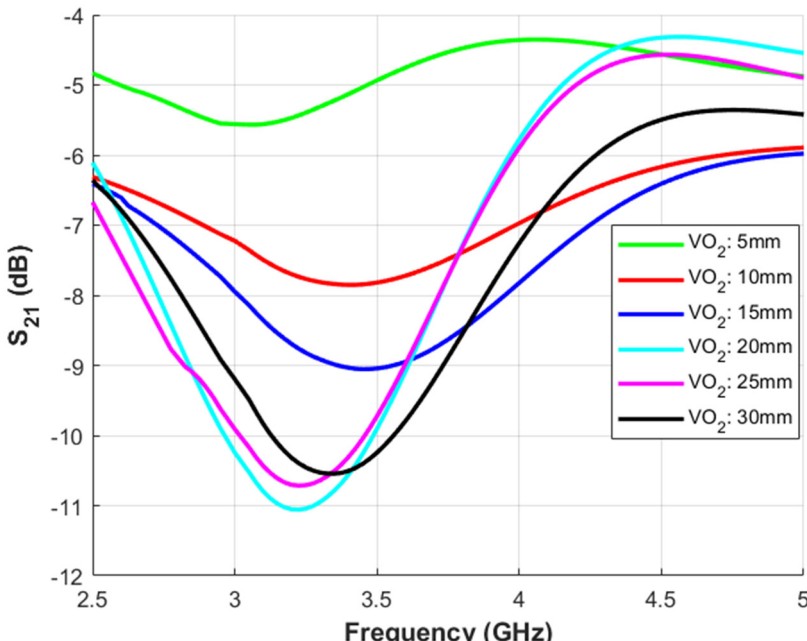

**Figure 11.** The graph of the measured transmission coefficient of the double diffracted dipole pattern on VO$_2$ over sapphire wafer with varying separation distance at L = 39 mm and W = 3 mm.

The magnitude of the transmission coefficient ($S_{21}$) for VO$_2$ double dipoles of spacings 25 and 30 mm compared to Cu is shown in Figure 12. The FSS response offers stable and identical transmission responses for both materials. From Figure 12, the VO$_2$ dipoles with 25 mm spacing at −10 dB provides 1 GHz bandwidth of stopband. This response makes it attractive as a filter in communication. The rejection bandwidth increases as the spacing is varied.

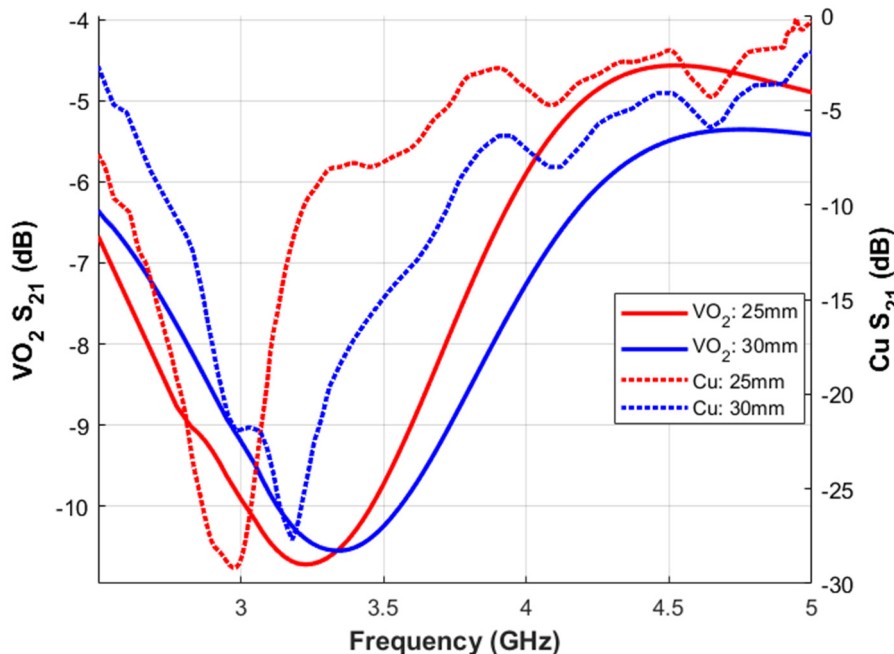

**Figure 12.** The graph of the measured transmission coefficient of VO$_2$ and Cu double dipoles for separations of 25 mm and 30 mm.

### 3.2.3. Full Dipole Array

The FSS shown in Figure 5a consists of a 3 × 5-unit cell arrangement at 10 GHz projected on the VO$_2$ film circular surface of 50.8 mm diameter. The unit cell is made of dipoles of length, $l(y)$ = 8.3 mm, width, $w(x)$ = 3 mm, with the interelement spacings of $dx(x)$ and $dy(y)$ as 7.5 and 12 mm, respectively. This design is selected to fit an array on the VO$_2$ surface. The dipole is used to demonstrate the proposed design of the FSS element in Figure 13. The proposed design employs variable state VO$_2$ and is supported by a sapphire substrate with a thickness, $h$, of 420 μm and a dielectric constant, $\varepsilon_r$, of 11. CST Microwave Studio software simulates the proposed FSS's unit cell. The off-state (below transition temperature) of VO$_2$ is modelled as "normal material" with a dielectric constant, $\varepsilon_r$, of 34 and an electrical conductivity of 1.2 S/m. On the other hand, the on-state (above transition temperature) of VO$_2$ is modelled as a lossy metal with an electrical conductivity of $1.2 \times 10^3$ S/m.

The frequency domain solver was used for the simulation because it is appropriate for the simulation of a highly resonant structure, as well as acknowledging the examination of the FSS performance with variations in incident angle and polarisation. For the port assignment, the periodic boundary condition is set in horizontal and vertical directions, and the z-direction is set to open boundary conditions. The CST full Floquet port is used to simulate the presented FSS as an infinite array in the CST. This setting also helps to reduce the simulation load. Figure 14 depicts the proposed FSS's simulated transmission response at the normal incident for both TE and TM polarisation.

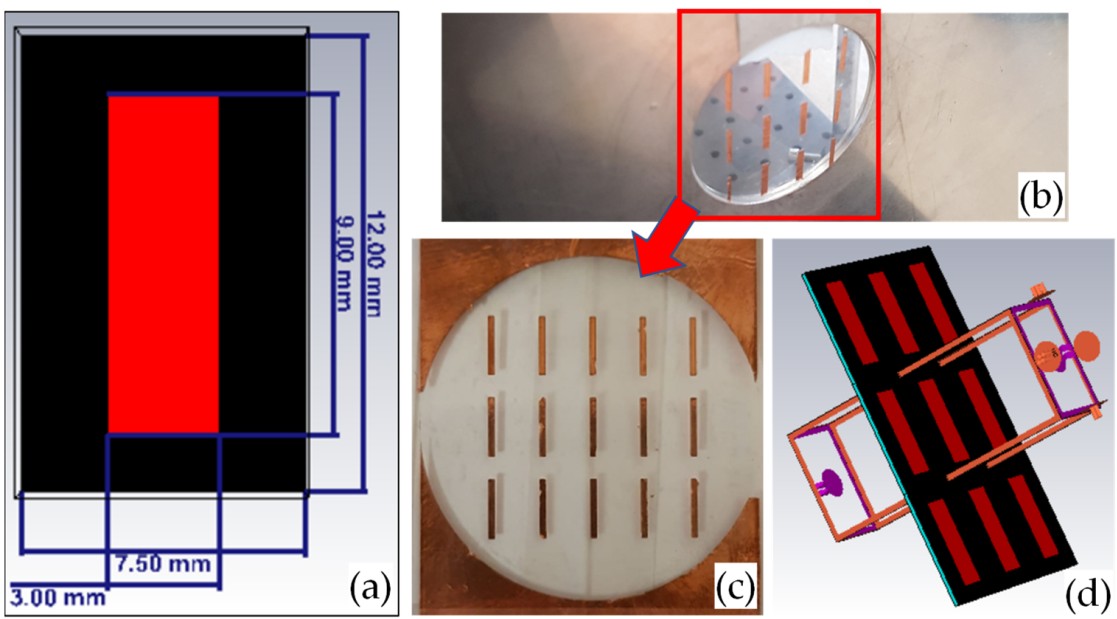

**Figure 13.** FSS design of: (**a**) VO$_2$ showing a hot state (red) modelled as a lossy metal with an electrical conductivity of $1.2 \times 10^3$ S/m and cold state (black) modelled as "normal material" with a dielectric constant, $\varepsilon_r$, of 34 and an electrical conductivity of 1.2 S/m in CST software; (**b**) Cu FSS structure on the sample holder; (**c**) Cu FSS array on a glass substrate; (**d**) Modelling FSS structure on sapphire in CST software.

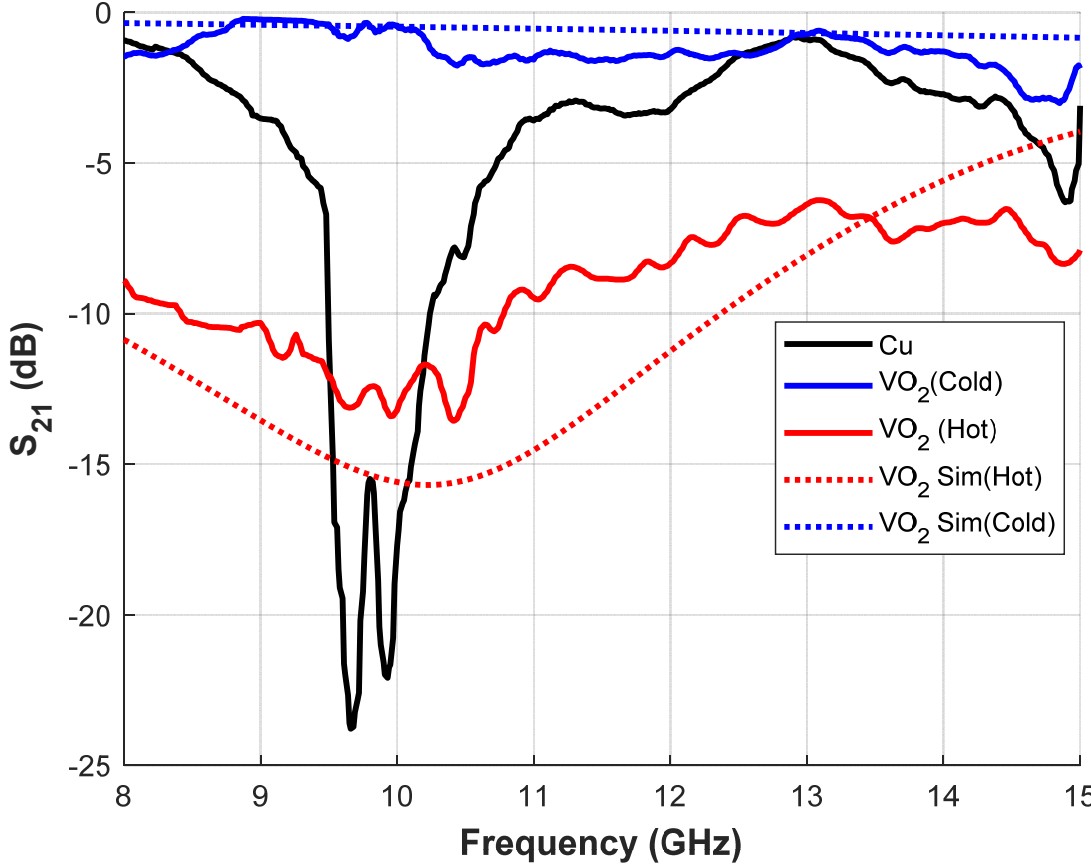

**Figure 14.** Graph of the measured and simulated transmission coefficient of the FSS dipole array.

The thermal nature of the VO$_2$ layer is considered in the simulation model by varying the electrical conductivity across the unit cell structure in CST software. Figure 13 shows the simulation model and fabricated Cu FSS structure of the 10 GHz image projected in Figure 5a. The transmission coefficient for the dipole FSS array is represented in Figure 14. In this graph, the measured and simulated VO$_2$ array responses are nearly the same as each other, with some slight variations. The resonant frequency for both the simulated and measured VO$_2$ array is 10.5 GHz, with the simulated response having a sharper transmission coefficient of 16 dB, while the measured response has a broader result of 13 dB across 9.5–10.5 GHz. On the other hand, the response of Cu FSS shows the transmission coefficients of $-24$ dB and $-22$ dB at 9.6 and 9.9 GHz of resonant frequency, respectively.

The discrepancy between Cu and VO$_2$, apart from the former's higher conductivity value, is due to heat expansion. The heat expansion is maintained enough to form dipoles on the VO$_2$ film, resulting in a filtering effect at 9.5–10.5 GHz; however, the heating still raises the global sample temperature even at the interelement spaces surrounding the dipole elements, which leads to a broader transmission curve and lowers the overall response. This temperature change is evident in the value between the lowest and highest S$_{21}$ values of VO$_2$ and Cu.

Figure 14 proves that VO$_2$ has the potential to be used in reconfigurable devices. By moving to lower frequencies, the unit cell's size increases, making it easier to contain the heat expansion. The cost and technology restrictions on the deposition of VO$_2$ thin film make producing larger FSS array designs more challenging.

Radar cross-section (RCS) measurements were carried out on a fabricated FSS Cu array reflector at 14 GHz. The unit cell properties are used as a reference to analyse and estimate VO$_2$ array response [4], as shown in Figure 15. The Cu simulation is matched to measurement results and compared to the VO$_2$ array response in CST studio.

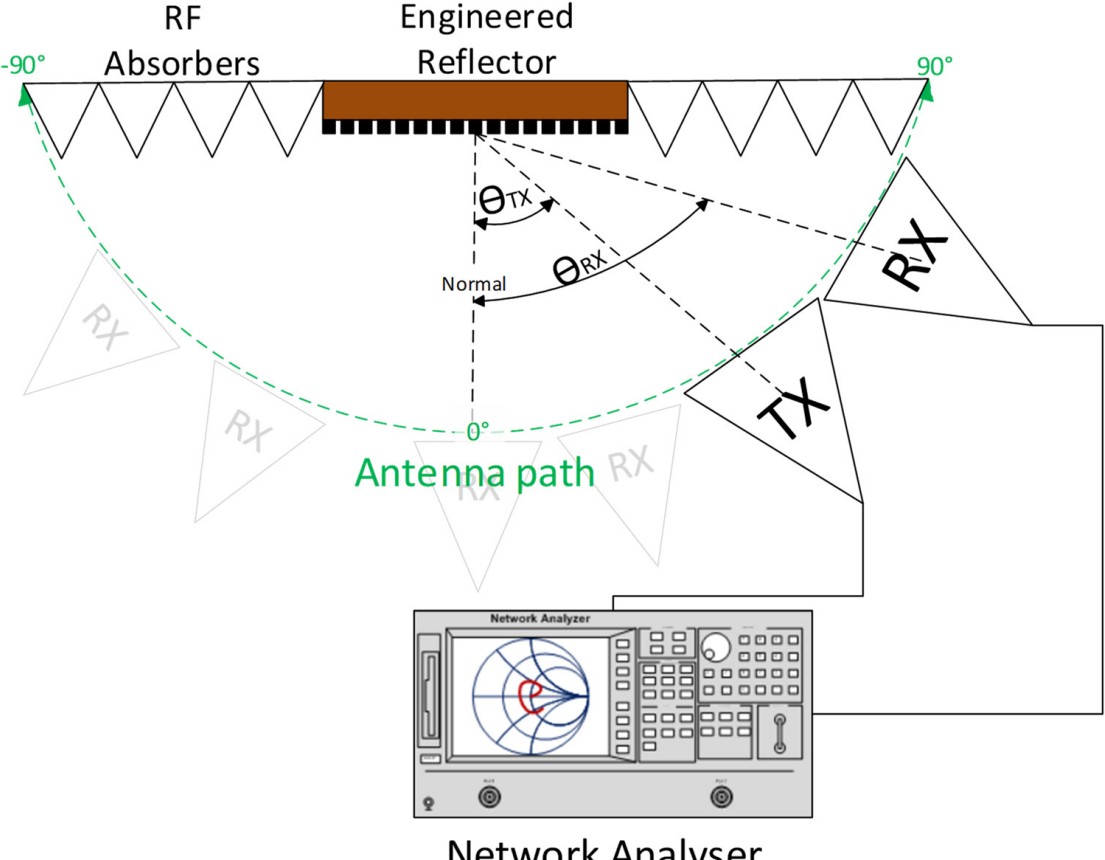

**Figure 15.** Bistatic RCS measurement setup with fixed transmitter (Tx).

The bistatic RCS for TE polarised waves at an incidence angle of $45°$ is obtained using a finite array with $10 \times 28$-unit cells with size of $18 \times 44$ cm$^2$, as illustrated in Figure 16. The RCS for Cu array reflects the incident signal to the impinging direction at $-31$ dB, while the VO$_2$ array in the on-state reflects at $-45$ dB. The discrepancy of the magnitude of reflection can be attributed partly to the conductor and dielectric losses exhibited by the array (the heated VO$_2$ is not a perfect electric conductor) and partly to design imperfections. It is highlighted that the angle of reflection remains unchanged.

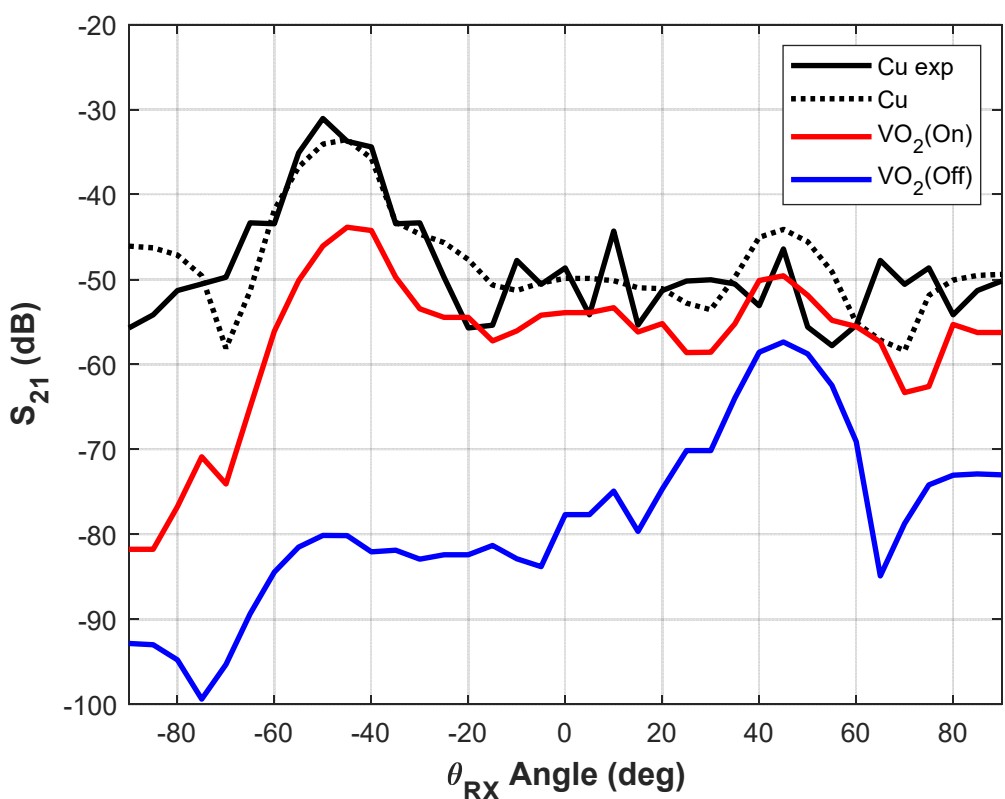

**Figure 16.** Graph of the bistatic RCS responses of the manufactured and evaluated FSS reflectors with fixed Tx at $-45°$.

## 4. Conclusions

A reconfigurable selective surface using the thermochromic property of VO$_2$ material for electromagnetic wave manipulation was proposed. The use of VO$_2$ film as an FSS array for reconfigurable spatial filtering is demonstrated. Using laser heating and a DOE lens, we have demonstrated that it can project FSS dipoles onto the VO$_2$ surface to produce frequency-selective surfaces via adequate heat management. The proposed technique will offer a solution for reconfigurable metamaterial applications.

Moreover, it has been shown that the transmission response of the VO$_2$ FSS is the same as that of a Cu FSS, particularly at lower frequencies with more giant FSS unit cells, where the dipole image has better thermal management to maintain the FSS periodicity. The results are further validated by simulation and a fabricated FSS prototype, which were in good agreement. Extension of the proposed solution to reflective surfaces can readily be made using the same design concepts with an adequate heat management scheme. Beyond the FSS application, the proposed concept is suitable for several other quasi-optical and quantum applications where polarisation-selective surfaces are required.

**Author Contributions:** Conceptualisation, D.E.A., G.G. and D.H.; methodology, Q.A., A.D. and C.R.; software, Q.A.; validation, Q.A., A.D. and C.R.; formal analysis, Q.A. and A.D.; investigation, Q.A., A.D. and C.R.; resources, D.E.A., G.G. and D.H.; data curation, Q.A.; writing—original draft preparation, Q.A.; writing—review and editing, Q.A., A.D., C.R., G.J., N.S., A.S., G.G., D.H. and D.E.A.; visualisation, A.D.; supervision, D.E.A., G.G. and D.H.; project administration, D.E.A., G.G. and D.H.; funding acquisition, D.E.A., G.G. and D.H. All authors have read and agreed to the published version of the manuscript.

**Funding:** This research was partly supported by the Defence and Security Accelerator project #ACC6003961 and the European H2020 Marie Skłodowska-Curie Individual Fellowship (MSCA-IF) grant #840854 (VisionRF).

**Data Availability Statement:** Not applicable.

**Conflicts of Interest:** The authors declare no conflict of interest.

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
