# Peer review of "Use of Thermochromic Properties of VO2 for Reconfigurable Frequency Selection"

_electronics, doi:10.3390/electronics11244099_

Round 1
Reviewer 1 Report
This work explored the thermochromic nature of vanadium dioxide (VO2). The authors demonstrated that it is possible to project FSS dipoles onto the VO2, while it also had the same transmission response as Cu FSS. I find this manuscript important for future metamaterial research. The topic falls within the scope of the Electronics journal. I recommend acceptance of this manuscript in its current form.
Author Response
We sincerely thank the Reviewer for the time taken to review this work and the recommendation for acceptance.

Reviewer 2 Report
In this work we use diffractive optical elements to form conductive patterns on a purely VO2 thin film and measure the transmission characteristics. The work investigates the transition behavior of VO2 to demonstrate an optically actuated unit cell in the 2.5 GHz to 5 GHz bands. I think paper is interesting I would propose some changes as follows:
1. Authors should justify usage of vanadium dioxide.
2. Authors should stress novelty of their work in comparison to others.
3. Authors are missing some recent articles in the field such as Spoof plasmons in corrugated transparent conducting oxides.
4. Authors should justify the usage of the proposed frequency range.
Author Response
Please see attached file with our modifications.

Reviewer 3 Report
1Using the phase changing property of VO2, a 13 reconfigurable metamaterial concept is achieved capable of manipulating electromagnetic waves 14 for different functionalities in this paper. To achieve this, diffractive optical elements (DOE) are used to project 15 conductive images on VO2 wafer for frequency selective surface (FSS) applications. The work is novel and interesting, and I believe it can be accepted for publication with the following modifications and additions:
11. The author gives the sample diagram of vanadium dioxide in Figure 3, but there is not much explanation in the manuscript, please add.
22. The format in the author's manuscript needs to be carefully checked. There should be a space between the number and the unit in the manuscript, such as "3GHz". Please modify it.
33. There is a rapid development in the aspects of metasurface based on VO2 , the following articles had better be cited. Dynamically tunable broadband absorber/reflector based on graphene and VO2 metamaterials Applied Optics 61 (7), 1646-1651
Author Response

(The authors gave the same response as above.)

Author Response

(The authors gave the same response as above.)

Round 2
Reviewer 4 Report
The authors have carefully responsed all the questions. I recommend acceptance of this manuscript in its current form.